# Glutaminase 2 is a novel negative regulator of small GTPase Rac1 and mediates p53 function in suppressing metastasis

Cen Zhang[1†], Juan Liu[1†], Yuhan Zhao[1], Xuetian Yue[1], Yu Zhu[1,2], Xiaolong Wang[1], Hao Wu[1], Felix Blanco[1], Shaohua Li[3], Gyan Bhanot[4], Bruce G Haffty[1], Wenwei Hu[1*], Zhaohui Feng[1*]

[1]Department of Radiation Oncology, Rutgers Cancer Institute of New Jersey, Rutgers, The State University of New Jersey, New Brunswick, United States; [2]Department of Neurosurgery, First Affiliated Hospital, Zhejiang University School of Medicine, Hangzhou, China; [3]Department of Surgery, Robert Wood Johnson Medical School, Rutgers, The State University of New Jersey, New Brunswick, United States; [4]Department of Molecular Biology, Biochemistry & Physics, Rutgers, The State University of New Jersey, Piscataway, United States

**Abstract** Glutaminase (GLS) isoenzymes GLS1 and GLS2 are key enzymes for glutamine metabolism. Interestingly, GLS1 and GLS2 display contrasting functions in tumorigenesis with elusive mechanism; GLS1 promotes tumorigenesis, whereas GLS2 exhibits a tumor-suppressive function. In this study, we found that GLS2 but not GLS1 binds to small GTPase Rac1 and inhibits its interaction with Rac1 activators guanine-nucleotide exchange factors, which in turn inhibits Rac1 to suppress cancer metastasis. This function of GLS2 is independent of GLS2 glutaminase activity. Furthermore, decreased GLS2 expression is associated with enhanced metastasis in human cancer. As a p53 target, GLS2 mediates p53's function in metastasis suppression through inhibiting Rac1. In summary, our results reveal that GLS2 is a novel negative regulator of Rac1, and uncover a novel function and mechanism whereby GLS2 suppresses metastasis. Our results also elucidate a novel mechanism that contributes to the contrasting functions of GLS1 and GLS2 in tumorigenesis.

*For correspondence: wh221@ cinj.rutgers.edu (WH); fengzh@ cinj.rutgers.edu (ZF)

[†]These authors contributed equally to this work

Competing interests: The authors declare that no competing interests exist.

## Introduction

Metabolic changes are a hallmark of cancer cells (*Berkers et al., 2013*; *Cairns et al., 2011*; *Ward and Thompson, 2012*). Increased glutamine metabolism (glutaminolysis) has been recognized as a key metabolic change in cancer cells, along with increased aerobic glycolysis (the Warburg effect) (*Berkers et al., 2013*; *Cairns et al., 2011*; *DeBerardinis et al., 2007*; *Hensley et al., 2013*; *Ward and Thompson, 2012*). Glutamine is the most abundant amino acid in human plasma (*Hensley et al., 2013*). Glutamine catabolism starts with the conversion of glutamine to glutamate, which is converted to α-ketoglutarate for further metabolism in the tricarboxylic acid (TCA) cycle. Recent studies have shown that increased glutamine metabolism plays a critical role in supporting the high proliferation and survival of cancer cells by providing pools of the TCA cycle intermediates, as well as the biosynthesis of proteins, lipids, and nucleotides (*Berkers et al., 2013*; *Cairns et al., 2011*; *DeBerardinis et al., 2007*; *Hensley et al., 2013*; *Ward and Thompson, 2012*).

Glutaminase (GLS) is the initial enzyme in glutamine metabolism, which catalyzes the hydrolysis of glutamine to glutamate in cells. Two genes encode glutaminases in human cells: GLS1 (also known

**eLife digest** Healthy cells in the body derive most of their energy from a sugar called glucose. However, cancer cells grow and divide much more rapidly than normal cells and so require larger amounts of energy to sustain themselves. Therefore, many cancer cells can alter their metabolism so that they can obtain more energy from a molecule called glutamine or other alternative sources.

Cancer cells obtain glutamine from the blood and use an enzyme called glutaminase to convert it into another type of molecule. Human cells produce two forms of glutaminase called GLS1 and GLS2. Even though both enzymes share many common features, they have different effects on cancer cells. GLS1 promotes tumor formation, while GLS2 has the opposite effect. However, it is not clear why these enzymes behave so differently.

Zhang, Liu et al. now investigate how GLS2 suppresses the progression of tumors. The experiments show that GLS2, but not GLS1, can directly bind to a protein called Rac1 that normally promotes the spread of tumor cells around the body. GLS1 inhibits the activity of Rac1, but this happens independently of the enzyme's glutaminase activity. Zhang, Liu et al. altered the levels of GLS2 in liver cancer cells and then injected these cells into mice. Cells that had low levels of GLS2 were able to spread and form tumors in distant sites like the lung. In contrast, smaller and fewer lung tumors were observed in mice that had been injected with cells that produced high levels of GLS2.

Zhang, Liu et al.'s findings reveal a new role for GLS2 that may help to explain why it affects tumor progression differently from GLS1. Further work is now needed to explore whether targeting Rac1 could be a potential therapy for cancers that have lost GLS2.

as kidney-type glutaminase), and GLS2 (also known as liver-type glutaminase). GLS1 and GLS2 proteins exhibit a high degree of amino acid sequence similarity, particularly in their glutaminase core domains. While GLS1 and GLS2 both function as glutaminase enzymes in glutamine metabolism, recent studies show that they have very different functions in tumorigenesis. GLS1 is ubiquitously expressed in various tissues, and its expression can be induced by the oncogene MYC (*Gao et al., 2009*). GLS1 is frequently activated and/or overexpressed in various types of cancer, including hepatocellular carcinoma (HCC) (*Gao et al., 2009*; *Thangavelu et al., 2012*; *Wang et al., 2010*; *Xiang et al., 2015*). GLS1 has been reported to promote tumorigenesis in different types of cancer, including HCC, which is mainly attributable to its glutaminase activity and role in promoting glutamine metabolism (*Gao et al., 2009*; *Thangavelu et al., 2012*; *Wang et al., 2010*; *Xiang et al., 2015*). By contrast, GLS2 is specifically expressed in only a few tissues, including the liver tissue. Recent studies including ours have shown that *GLS2* is a novel target gene of the tumor suppressor p53. GLS2 is transcriptionally up-regulated by p53 and mediates p53's regulation of mitochondrial function and anti-oxidant defense in cells (*Hu et al., 2010*; *Suzuki et al., 2010*). Considering the critical role of p53 and its pathway in tumor suppression, the identification of *GLS2* as a p53 target gene strongly suggests a potentially important role of GLS2 in tumor suppression. Recent studies have shown that, in contrast to the tumorigenic effect of GLS1, GLS2 displays a tumor suppressive function (*Hu et al., 2010*; *Liu et al., 2014a*; *Suzuki et al., 2010*). GLS2 expression is frequently reduced in HCC (*Hu et al., 2010*; *Liu et al., 2014a*; *Suzuki et al., 2010*; *Xiang et al., 2015*). Ectopic expression of GLS2 greatly inhibited the growth and colony formation of HCC cells in vitro and the growth of HCC xenograft tumors in vivo (*Hu et al., 2010*; *Liu et al., 2014a*; *Suzuki et al., 2010*). Given that GLS1 and GLS2 both function as glutaminase enzymes, the mechanisms underlying their contrasting roles in tumorigenesis remain unclear.

In this study, immunoprecipitation (IP) followed by liquid chromatography-tandem mass spectrometry (LC/MC-MS) analysis was employed to screen for potential proteins interacting with GLS2. The small GTPase Rac1 was identified as a novel binding protein for GLS2. Rac1 cycles between inactive guanosine 5'-diphosphate (GDP)-bound and active guanosine 5'-triphosphate (GTP)-bound forms in cells, and regulates a diverse array of cellular events, including actin dynamics. The Rac1 signaling is frequently activated in various types of cancer, in which it plays a critical role in promoting migration, invasion and metastasis of cancer cells (*Bid et al., 2013*; *Heasman and Ridley, 2008*). We

found that GLS2 binds to Rac1, and inhibits the interaction of Rac1 with its guanine-nucleotide exchange factors (GEFs) such as Tiam1 and VAV1, which would normally activate Rac1. Thus, GLS2 inhibits Rac1 activity, which in turn inhibits migration, invasion and metastasis of cancer cells. This function of GLS2 requires the C-terminus of GLS2 and is independent of its glutaminase activity. In contrast, GLS1 does not interact with Rac1 to inhibit Rac1 activity, and consequently, cannot inhibit cancer metastasis via this pathway. p53 plays a pivotal role in suppressing cancer metastasis, but its underlying mechanism is not fully understood (*Muller et al., 2011*; *Vousden and Prives, 2009*). Our results further show that, as a direct downstream target of p53, GLS2 mediates p53's function in metastasis suppression through inhibiting the Rac1 signaling. Taken together, our results demonstrated that GLS2 is a novel negative regulator of Rac1, and plays a critical role in suppression of metastasis through its negative regulation of Rac1 activity. Our results also revealed that GLS2 plays an important role in mediating the function of p53 in suppression of cancer metastasis.

## Results

### Rac1 is a novel GLS2 interacting protein

GLS2 was reported to interact with several proteins although the biological functions of these interactions remain unclear (*Boisguerin et al., 2004*; *Olalla et al., 2001*). These findings raised the possibility that GLS2 may exert its function in tumor suppression through its interactions with other proteins. Herein, we screened for potential GLS2-interacting proteins in human HCC Huh-1 cells stably transduced with pLPCX-GLS2-Flag retroviral vectors to express GLS2-Flag and control cells transduced with control vectors. Co-IP assays using an anti-Flag antibody followed by LC-MS/MS assays were employed. These assays identified the small GTPase Rac1 as a potential GLS2 interacting protein (*Figure 1A*). Rac1 is frequently activated or overexpressed in various types of cancer, including HCC, and has been reported to play a critical role in promoting cancer cell migration, invasion and metastasis mainly through its regulation of actin dynamics (*Bid et al., 2013*; *Heasman and Ridley, 2008*).

The interaction between GLS2 and Rac1 was confirmed by co-IP followed by western blot assays in Huh-1 cells co-transduced with the vectors expressing GLS2-Flag and Myc-Rac1, respectively. GLS2-Flag was co-precipitated by the anit-Myc antibody, and Myc-Rac1 was co-precipitated by the anti-Flag antibody, indicating that GLS2-Flag interacted with Myc-Rac1 in cells (*Figure 1B*). In contrast, no interaction was observed between GLS1-Flag and Myc-Rac1 in Huh-1 cells (*Figure 1C*). The interaction between endogenous Rac1 with endogenous GLS2 but not GLS1 was also observed in Huh-1 and HepG2 cells (*Figure 1D*).

To identify the domain of GLS2 that interacts with Rac1, three Flag-tagged deletion mutants of GLS2 were constructed, including ΔN163 (deletion of N-terminal amino acids (aa) 1–163), ΔC139 (deletion of C-terminal aa 464–602), and C139 (C-terminal aa 464–602 only) (*Figure 1E*). Co-IP assays in Huh-1 cells showed that the GLS2-ΔN163 and GLS2-C139, but not GLS2-ΔC139, interacted with Myc-Rac1, indicating that the C-terminus of GLS2 is necessary and sufficient for the interaction between GLS2 and Rac1 (*Figure 1F*). Since the C-terminus of GLS2 (GLS2-C139) does not contain the glutaminase core domain, which encodes the glutaminase catalytic domain (*Figure 1E*) and hence lacks glutaminase activity (*Figure 1G*), these results demonstrate that the binding of GLS2 to Rac1 is independent of its glutaminase activity.

### GLS2 interacts with Rac1-GDP and inhibits the Rac1 activity

As a molecular switch, Rac1 cycles between inactive GDP-bound and active GTP-bound forms in cells (*Bid et al., 2013*; *Heasman and Ridley, 2008*; *Raftopoulou and Hall, 2004*). It is well known that constitutively active mutant Rac1 (CA Rac1-G12V) exists constitutively in the GTP-bound form in cells, whereas the dominant negative Rac1 mutant (DN Rac1-T17N) exists constitutively in the GDP-bound form in cells (*Feig, 1999*; *Ridley et al., 1992*). Rac1-GDP and Rac1-GTP display different conformations and interact with different proteins. For instance, GEFs specifically bind to Rac1-GDP to catalyze the exchange of GDP to GTP and thereby activate Rac1, whereas GAPs (GTPase-activating proteins) specifically bind to Rac1-GTP to hydrolyze GTP and thereby inactivate Rac1 (*Cherfils and Zeghouf, 2013*; *Vetter and Wittinghofer, 2001*).

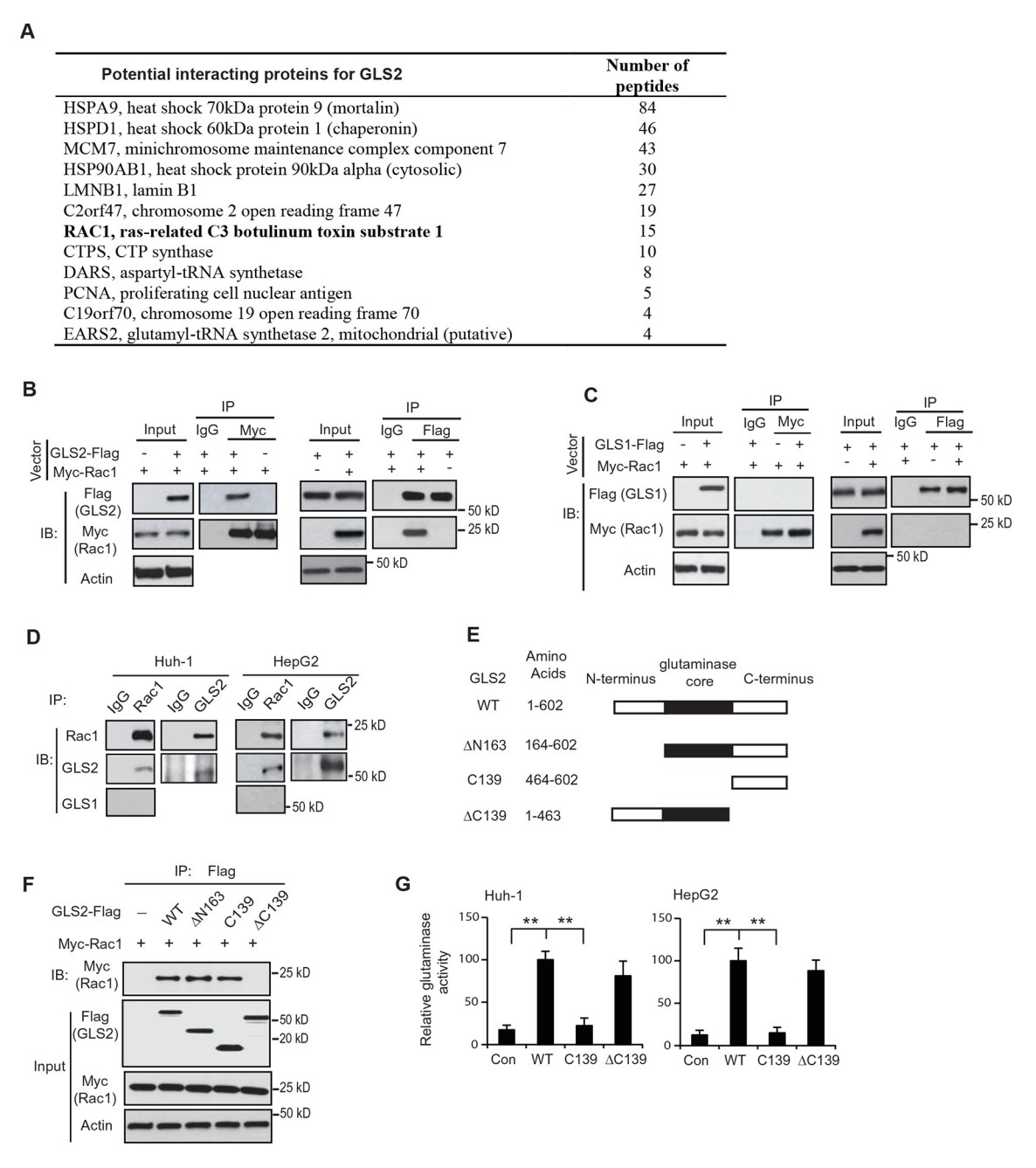

**Figure 1.** Rac1 is a novel interacting protein for GLS2. (**A**) The potential GLS2-interacting proteins identified by co-IP followed by LC-MS/MS analysis. Huh-1 cells expressing GLS2-Flag or cells transduced with control vectors were used for co-IP with the anti-Flag antibody followed by LC-MS/MS analysis. The potential GLS2 interacting proteins are listed with the number of peptides identified by LC-MS/MS analysis. (**B**) GLS2-Flag interacted with Myc-Rac1 in cells. Huh-1 cells were transduced with Myc-Rac1, GLS2-Flag and control vectors as indicated for co-IP assays using the anti-Myc (left panels) and anti-Flag antibodies (right panels), respectively. (**C**) GLS1-Flag did not interact with Myc-Rac1 in cells. Huh-1 cells were transduced with Myc-Rac1 and GLS1-Flag vectors for co-IP assays using the anti-Myc (left panels) and anti-Flag antibodies (right panels), respectively. (**D**) Endogenous GLS2 but not GLS1 interacted with endogenous Rac1 in Huh-1 and HepG2 cells detected by co-IP assays. (**E**) Schematic representation of vectors expressing Flag-tagged WT or serial deletion mutants of GLS2. (**F**) The C-terminus of GLS2, GLS2-C139, is necessary and sufficient for GLS2 to interact with Rac1. Huh-1 cells were transduced with WT or different mutant GLS2-Flag vectors listed in *Figure 1E* together with Myc-Rac1 vectors for co-IP assays. (**G**) The relative glutaminase activities of WT and different mutant GLS2. Huh-1 and HepG2 cells were transduced with WT and different mutant GLS2 vectors. The relative glutaminase activities in cells transduced with WT GLS2 vectors were designated as 100. **: $p<0.001$. Student's *t*-test. GLS, glutaminase; IB: immunoblot; IP, immunoprecipitation; LC/MC-MS, liquid chromatography-tandem mass spectrometry; WT, wild-type.

To investigate the biological function of the interaction between GLS2 and Rac1, Huh-1 cells were co-transfected with GLS2-Flag vectors and CA Myc-Rac1-G12V or DN Myc-Rac1-T17N vectors for co-IP assays. We found that GLS2-Flag preferentially bound to Myc-Rac1-T17N but not Myc-Rac1-G12V (*Figure 2A*). To confirm this result, lysates from Huh-1 cells co-transduced with GLS2-Flag and Rac1-Myc were pretreated with GDP or GTPγS (a non-hydrolyzable analog of GTP) to convert Rac1 in cell lysates into Rac1-GDP or Rac1-GTP form as previously described (*Castillo-Lluva et al., 2010*; *Fukata et al., 2002*). Co-IP assays showed that GLS2-Flag preferentially bound to Myc-Rac1 in cell lysates pretreated with GDP but not GTPγS (*Figure 2B*). These results showed that GLS2 preferentially binds to Rac1-GDP, suggesting that GLS2 is involved in regulating Rac1 activity.

We further investigated whether GLS2 inhibits the Rac1 activity in different human HCC cell lines, including Huh-1 and HepG2 (p53 wild type; WT), Hep3B (p53-null) and Huh-7 cells (p53 mutant). PAK1 (p21-activated kinase 1) is a critical Rac1 effector protein. It has been well established that the p21-binding domain of PAK1 binds specifically to the Rac1-GTP but not Rac1-GDP in cells (*Galic et al., 2014*; *Hayashi-Takagi et al., 2010*; *Palacios et al., 2002*). Based on this fact, the GST-p21-binding domain of PAK1 pull-down assays have been widely used to measure the levels of Rac1-GTP in cells as a standard method to analyze the Rac1 activity in cells (*Galic et al., 2014*; *Hayashi-Takagi et al., 2010*; *Palacios et al., 2002*). We found that ectopic expression of GLS2-Flag greatly decreased the levels of Rac1-GTP measured by p21-binding domain of PAK1 pull-down assays in Huh-1 and HepG2 cells (*Figure 2C*), as well as Hep3B and Huh-7 cells (*Figure 2C* and *Figure 2—figure supplement 1A*). In contrast, the expression of GLS2-Flag did not affect the levels of total Rac1 protein in these cells measured by western blot assays (*Figure 2C* and *Figure 2—figure supplement 1A*), indicating that GLS2 inhibits Rac1 activity. Consistently, knockdown of GLS2 greatly increased the levels of Rac1-GTP but not total Rac1 in Huh-1 and HepG2 cells (*Figure 2D*) as well as in Hep3B and Huh-7 cells (*Figure 2D* and *Figure 2—figure supplement 1B,C*). The endogenous levels of GLS2 in these HCC cells and the levels of GLS2-Flag in HCC cells transduced with GLS2-Flag expression vectors were shown in *Figure 2—figure supplement 2A–D*. It has been known that Rac1 binds to PAK1 and results in the auto-phosphorylation of PAK1 at multiple sites, including Ser199/204, leading to PAK1 activation (*Chong et al., 2001*; *Heasman and Ridley, 2008*). Therefore, we further investigated the effect of GLS2 on Rac1 activity by detecting the Ser199/204 phosphorylation of PAK1 in cells. Ectopic expression of GLS2-Flag in Huh-1 and HepG2 cells greatly reduced Ser199/204 phosphorylation of PAK1 (*Figure 2E*), whereas GLS2 knockdown enhanced Ser199/204 phosphorylation of PAK1 (*Figure 2F*), which further indicates that GLS2 inhibits the Rac1 activity in HCC cells. In contrast, ectopic GLS1 expression or GLS1 knockdown did not affect the Rac1 activity in HCC cells (*Figure 2—figure supplement 1D,E*).

Consistent with WT GLS2, the C-terminus of GLS2, GLS2-C139, specifically bound to the DN Rac1-T17N but not CA Rac1-G12V in Huh-1 cells (*Figure 2G*). Furthermore, ectopic expression of GLS2-C139 greatly inhibited the Rac1 activity in Huh-1 cells (*Figure 2H*). In contrast, GLS2-ΔC139, which did not bind to Rac1 (*Figure 1F*), failed to inhibit the Rac1 activity (*Figure 2H*). This result indicates that the interaction between GLS2 and Rac1 is critical for GLS2 to inhibit the Rac1 activity. Furthermore, the function of GLS2 in binding to and inhibiting Rac1 is independent of its glutaminase activity since the C-terminus of GLS2 (GLS2-C139) lacks the glutaminase activity (*Figure 1G*). Collectively, these results revealed that GLS2 is a novel negative regulator of the Rac1 signaling; GLS2 inhibits the Rac1 activity through its interaction with Rac1-GDP, and furthermore, this function of GLS2 requires the C-terminus of GLS2 and is independent of GLS2 glutaminase activity.

## GLS2 inhibits the interaction of Rac1-GDP with Rac1 GEFs to negatively regulate Rac1

We further investigated the mechanism by which GLS2 inhibits Rac1. Rac1 was reported to interact with other proteins through its Switch I & II regions or its C-terminus, which contains the protein transduction domain and GTPase C-termini (*van Hennik et al., 2003*; *Vetter and Wittinghofer, 2001*) (*Figure 3A*). To examine the domain involved in the interaction of Rac1 with GLS2, different Myc-tagged deletion mutants of Rac1 were constructed, including the ΔC33 (deletion of aa 160–192), the ΔN29 (deletion of aa 1–29), and the ΔSwitch (deletion of aa 30–74) (*Figure 3A*). Co-IP assays showed that the Rac1-ΔC33 and Rac1-ΔN29, but not Rac1-ΔSwitch, interacted with GLS2-Flag (*Figure 3B*), indicating that the Switch I & II regions are necessary for Rac1 to interact with GLS2.

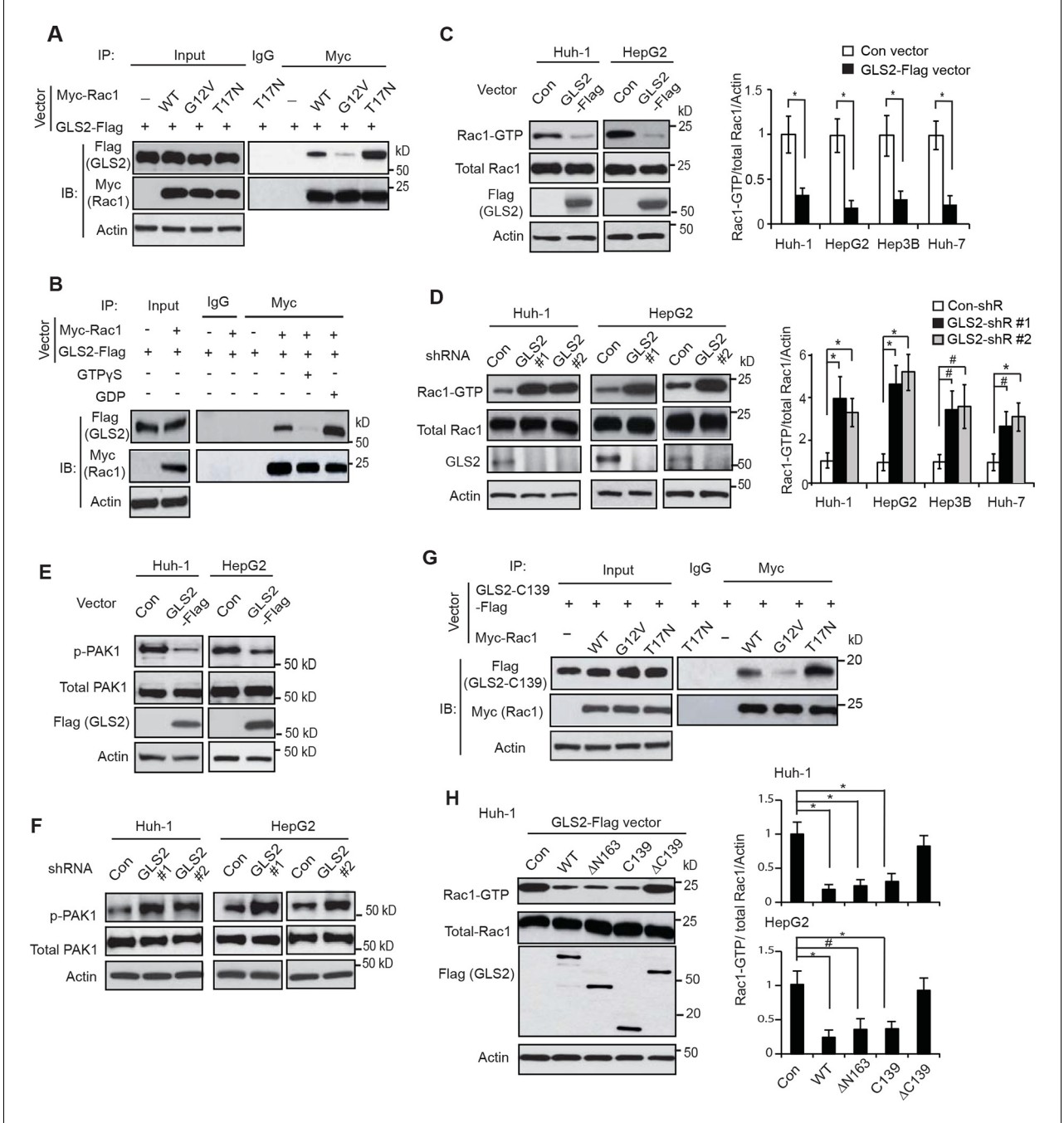

**Figure 2.** GLS2 interacts with Rac1-GDP and negatively regulates the Rac1 activity. (**A**) GLS2-Flag preferentially interacted with the DN Myc-Rac1-T17N but not the CA Myc-Rac1-G12V in Huh-1 cells. Cells were transduced with GLS2-Flag vectors together with Rac1-T17N or Rac1-G12V vectors for co-IP assays. (**B**) GLS2-Flag preferentially bound to Rac1-GDP but not Rac1-GTP in cell lysates. Cell lysates from Huh-1 cells transduced with vectors expressing Myc-Rac1 and GLS2-Flag were pretreated with GDP or GTPγS to convert Rac1 into Rac1-GDP or Rac1-GTP form before co-IP assays. (**C**) Ectopic expression of GLS2 inhibited Rac1 activities represented by decreased levels of Rac1-GTP in HCC cells measured by the GST-p21-binding domain of PAK1 pull-down assays. Left panels: Represented results of Rac1 activity analysis in Huh-1 and HepG2 cells. Right panels: relative Rac1-GTP/ total Rac1/Actin levels in Huh-1, HepG2, Hep3B and Huh-7 cells. Data present mean ± SD (n=3). *$p<0.01$; Student's $t$-test. (**D**) Knockdown of GLS2 by shRNA vectors increased Rac1 activities in HCC cells. Left panels: Represented results of Rac1 activity analysis in Huh-1 and HepG2 cells. Right panels: relative Rac1-GTP/total Rac1 /Actin levels in Huh-1, HepG2, Hep3B and Huh-7 cells. Data present mean ± SD (n=3). *$p<0.01$; #$p<0.05$; Student's $t$-test. (**E**) Ectopic expression of GLS2-Flag decreased the levels of p-PAK1 at Ser199/204 in Huh-1 and HepG2 cells. (**F**) Knockdown of GLS2 by shRNA vectors increased the levels of p-PAK1 at Ser199/204 in Huh-1 and HepG2 cells. (**G**) The C-terminus of GLS2, GLS2-C139, interacted with DN Myc-Rac1-T17N but not CA Myc-Rac1-G12V in Huh-1 cells detected by co-IP assays. (**H**) The C-terminus of GLS2, GLS2-C139, inhibited the Rac1 activity in Huh-1 and

*Figure 2 continued on next page*

*Figure 2 continued*

HepG2 cells. Left panels: Represented results of Rac1 activity analysis in Huh-1 cells transduced with different GLS2-Flag vectors. Right panels: relative Rac1-GTP/total Rac1/Actin levels in Huh-1 and HepG2. Data present mean ± SD (n=3). *$p<0.01$; #$p<0.05$; Student's *t*-test. GDP, guanosine 5'-diphosphate; GLS, glutaminase; GTP, guanosine 5'-triphosphate; HCC, hepatocellular carcinoma; IP, immunoprecipitation; shRNA, short hairpin RNA; WT, wild type.
The following figure supplements are available for figure 2:

**Figure supplement 1.** GLS2 inhibits Rac1 activity in HCC cells.
**Figure supplement 2.** The expression of endogenous GLS2 and exogenous GLS2-Flag in HCC cells.

It has been well-established that Rac1 GEFs can specifically bind to Rac1-GDP through the Switch I & II regions to catalyze the exchange of GDP to GTP to activate Rac1 (*Rossman et al., 2005*; *Vetter and Wittinghofer, 2001*). Tiam1 and VAV1 are two most common and critical GEFs of Rac1 (*Heo et al., 2005*; *Worthylake et al., 2000*). Consistent with previous reports, ectopically expressed Tiam1-HA and VAV1-HA specifically bound to Rac1-GDP (shown by their preferential interactions with Rac1-T17N but not Rac1-G12V; *Figure 3—figure supplement 1A,B*) through the Switch I & II regions (*Figure 3C*), leading to the activation of Rac1 in Huh-1 cells (*Figure 3D*). Since both GLS2 and Rac1 GEFs, such as Tiam1 and VAV1, bind to Rac1-GDP through the Switch I & II regions, this raised the possibility that GLS2 may inhibit Rac1 activity through competing with Rac1 GEFs for the Switch I & II regions of Rac1-GDP. To test this possibility, co-IP assays were performed in Huh-1 cells co-transduced with DN Myc-Rac1-T17N vectors and Tiam1-HA or VAV1-HA vectors, as well as increasing amount of vectors expressing GLS2-Flag. Increasing amount of GLS2-Flag resulted in a progressive reduction of Tiam1-HA or VAV1-HA bound to Myc-Rac1-T17N in cells (*Figure 3E,F*). Consistently transducing Huh-1 and HepG2 cells with increasing amount of GLS2-Flag vectors resulted in a progressive reduction of endogenous Tiam1 and VAV1 bound to endogenous Rac1 (*Figure 3G*). Furthermore, knockdown of endogenous GLS2 greatly promoted the interaction of endogenous Tiam1 and VAV-1 with endogenous Rac1 in Huh-1 and HepG2 cells (*Figure 3H*). These results suggest that GLS2 inhibits the Rac1 activation by interacting with Rac1-GDP to block its interaction with Rac1 GEFs, such as Tiam1 and VAV1.

## GLS2 inhibits migration and invasion of HCC cells through negative regulation of Rac1

GLS2 expression is frequently diminished in human HCC (*Hu et al., 2010*; *Liu et al., 2014a*; *Suzuki et al., 2010*; *Xiang et al., 2015*). However, its role in HCC, especially HCC metastasis, is poorly understood. The malignancy and poor prognosis of HCC has been related to the high metastatic characteristic of HCC (*El-Serag and Rudolph, 2007*; *Tang, 2001*). Currently, the mechanism underlying HCC metastasis is not well-understood. Rac1 is frequently activated or overexpressed in various types of cancer, including HCC, which plays a critical role in promoting cancer cell migration, invasion and metastasis (*Bid et al., 2013*; *Heasman and Ridley, 2008*). As shown in *Figure 4—figure supplement 1A–C*, ectopic expression of CA Myc-Rac1-G12V greatly promoted migration and invasion of Huh-1 and HepG2 cells as determined by trans-well assays, whereas expression of DN Myc-Rac1-T17N greatly inhibited migration and invasion of these cells. Therefore, our findings that GLS2 binds to and inhibits Rac1 raised the possibility that GLS2 may play an important role in suppressing cancer metastasis.

Here, we investigated the effects of GLS2 on the abilities of migration and invasion of different HCC cells, including Huh-1, HepG2, Hep3B and Huh7 cells, by using chamber trans-well assays. Cells were seeded into the upper chamber containing serum-free medium without or with matrigel for migration and invasion assays, respectively. Compared with cells transduced with control vectors, ectopic expression of GLS2 by GLS2-Flag retroviral vectors greatly reduced the migration and invasion of above-mentioned different HCC cells (*Figure 4A,B*). Furthermore, knockdown of GLS2 by short hairpin RNA (shRNA) vectors greatly promoted the migration and invasion of these cells (*Figure 4C,D*). Serum-free medium was used in the upper chamber to minimize the effect of GLS2 on cell proliferation in the trans-well assays. As shown in *Figure 4—figure supplement 2*, no

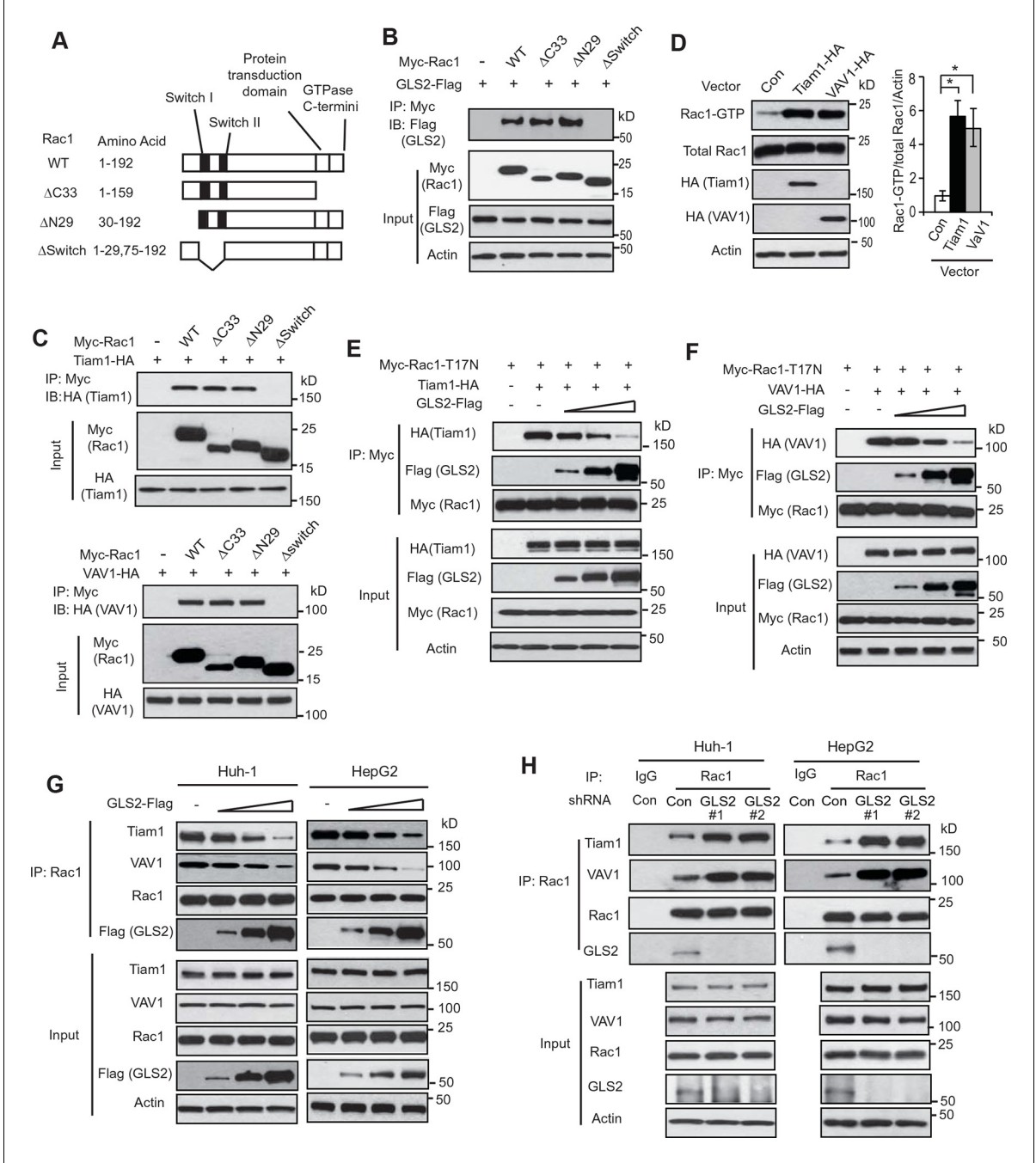

**Figure 3.** GLS2 inhibits the interaction of Rac1-GDP with Tiam1 and VAV1. (**A**) Schematic representation of Rac1 deletion mutants. Myc-tagged vectors expressing WT Rac1 or serial deletion mutants were constructed. (**B**) GLS2 bound to Rac1 through its Switch I & II regions. Huh-1 cells were transduced with different Myc-Rac1 vectors listed in *Figure 3A* together with GLS2-Flag vectors for co-IP assays. (**C**) Tiam1 (upper panels) and VAV1 (lower panels) bound to Rac1 through its Switch I & II regions. Huh-1 cells were transduced with different Myc-Rac1 vectors listed in *Figure 3A* together with vectors expressing Tiam1-HA or VAV1-HA for co-IP assays. (**D**) Ectopic expression of Tiam1 and VAV1 activated Rac1 in cells. Huh-1 cells were transduced with vectors expressing Tiam1-HA and VAV1-HA, respectively, and Rac1 activity was analyzed. Data are presented as mean ± SD (n=3). *$p<0.01$; Student's *t*-test. (**E, F**) Ectopic expression of GLS2 inhibited the interaction of DN Myc-Rac1-T17N with ectopic Tiam1-HA (**E**) and VAV1-HA (**F**) in cells. Huh-1 cells were transduced with Myc-Rac1-T17N vectors and Tiam1-HA (**E**) or VAV1-HA vectors (**F**) (2 µg), together with increasing amount of GLS2-Flag vectors (1, 3, 6 µg) for co-IP assays. (**G**) Ectopic expression of GLS2 inhibited the interaction of endogenous Rac1 with endogenous Tiam1 and VAV1 in cells. Huh-1 and HepG2 cells were transduced with increasing amount of GLS2-Flag expression vectors (1, 3, 6 µg) for co-IP assays. (**H**) Knockdown of endogenous GLS2 by shRNA vectors in Huh-1 and HepG2 cells promoted the interaction of endogenous Rac1 with endogenous Tiam1 and VAV1 as measured by co-IP assays. GLS, glutaminase; GTP, guanosine 5′-triphosphate; IP, immunoprecipitation; shRNA, short hairpin RNA; WT, wild type.

*Figure 3 continued on next page*

*Figure 3 continued*

The following figure supplement is available for figure 3:

**Figure supplement 1.** Tiam1 and VAV1 preferentially bind to Rac1-GDP.

significant difference in the viability and number of these cells among different groups was observed after being cultured in serum-free medium for 36 hr at the end of trans-well assays. Contrary to the role of GLS2 in suppressing migration and invasion, ectopic expression of GLS1-Flag significantly promoted the migration and invasion of Huh-1 and HepG2 cells (*Figure 4E*), whereas knockdown of endogenous GLS1 significantly reduced the migration and invasion of these cells (*Figure 4F*).

We further investigated whether GLS2 inhibits migration and invasion of HCC cells through its negative regulation of Rac1. Ectopic expression of the DN Myc-Rac1-T17N significantly reduced the migration and invasion of the above-mentioned four different HCC cells (*Figure 4G,H*). Notably, DN Myc-Rac1-T17N largely abolished the promoting effects of GLS2 knockdown on migration and invasion of these cells (*Figure 4G,H*). Consistently, knockdown of endogenous Rac1 significantly reduced the migration and invasion of Huh-1 and HepG2 cells, and, furthermore, largely abolished the promoting effects of GLS2 knockdown on migration and invasion (*Figure 4—figure supplement 3A–C*). No significant difference in the viability and number of these cells among different groups was observed after being cultured in serum-free medium for 36 hr at the end of trans-well assays (*Figure 4—figure supplement 3D*).

Consistent with WT GLS2, ectopic expression of the C-terminus of GLS2, GLS2-C139, which interacted with Rac1-GDP and inhibited the Rac1 activity (*Figure 2G,H*), greatly inhibited the migration and invasion of Huh-1 and HepG2 cells (*Figure 4I,J*). In contrast, deletion of the C-terminus of GLS2 (GLS2-ΔC139), which resulted in the loss of GLS2's ability to interact with Rac1 and inhibit the Rac1 activity (*Figure 2G,H*), largely abolished the ability of GLS2 to inhibit the migration and invasion of Huh-1 and HepG2 cells (*Figure 4I,J*). Taken together, these results demonstrate that the negative regulation of the Rac1 activity by GLS2 is crucial for GLS2 to inhibit the migration and invasion of cancer cells, and furthermore, this function of GLS2 requires the C-terminus of GLS2 and is independent of the glutaminase activity of GLS2.

## GLS2 inhibits lung metastasis of HCC cells in vivo through negative regulation of Rac1

Lung metastasis is the most frequently observed distant metastasis in HCC patients (*Kitano et al., 2012*; *Uka et al., 2007*). We investigated the effect of GLS2 on metastasis in vivo by employing the lung metastasis model in mice. Huh-1 and HepG2 cells with ectopic GLS2-Flag expression or GLS2 knockdown and their control cells were transduced with luciferase-expressing lentiviral vectors and injected into BALB/c athymic nude mice via the tail vein. The metastasis of HCC cells to lung was monitored by in vivo bioluminescence imaging. Bioluminescence imaging results showed that ectopic expression of GLS2-Flag in both Huh-1 and HepG2 cells significantly inhibited lung metastasis (*Figure 5A*). Histological analysis confirmed that mice injected with cells with ectopic GLS2-Flag expression had fewer and smaller metastatic tumors in the lung (*Figure 5B*). Furthermore, knockdown of endogenous GLS2 led to significantly increased lung metastasis of both Huh-1 and HepG2 cells analyzed by in vivo imaging and histological analysis, respectively (*Figure 5C,D*).

We further investigated whether inhibition of Rac1 mediates GLS2's function in suppression of lung metastasis of HCC cells in vivo. As shown in *Figure 5E,F*, ectopic expression of the DN Rac1-T17N greatly reduced lung metastasis of Huh-1 and HepG2 cells in nude mice. Notably, The DN Rac1-T17N largely abolished the promoting effects of GLS2 knockdown on lung metastasis of Huh-1 and HepG2 cells. These results together suggest that GLS2 inhibits cancer metastasis through its down-regulation of the Rac1 activity.

## The decreased GLS2 expression is associated with enhanced human HCC metastasis

Our results from cancer cell migration and invasion assays as well as lung metastasis models clearly showed that GLS2 inhibited metastasis of different human HCC cells, which strongly suggests that

eLIFEResearch article

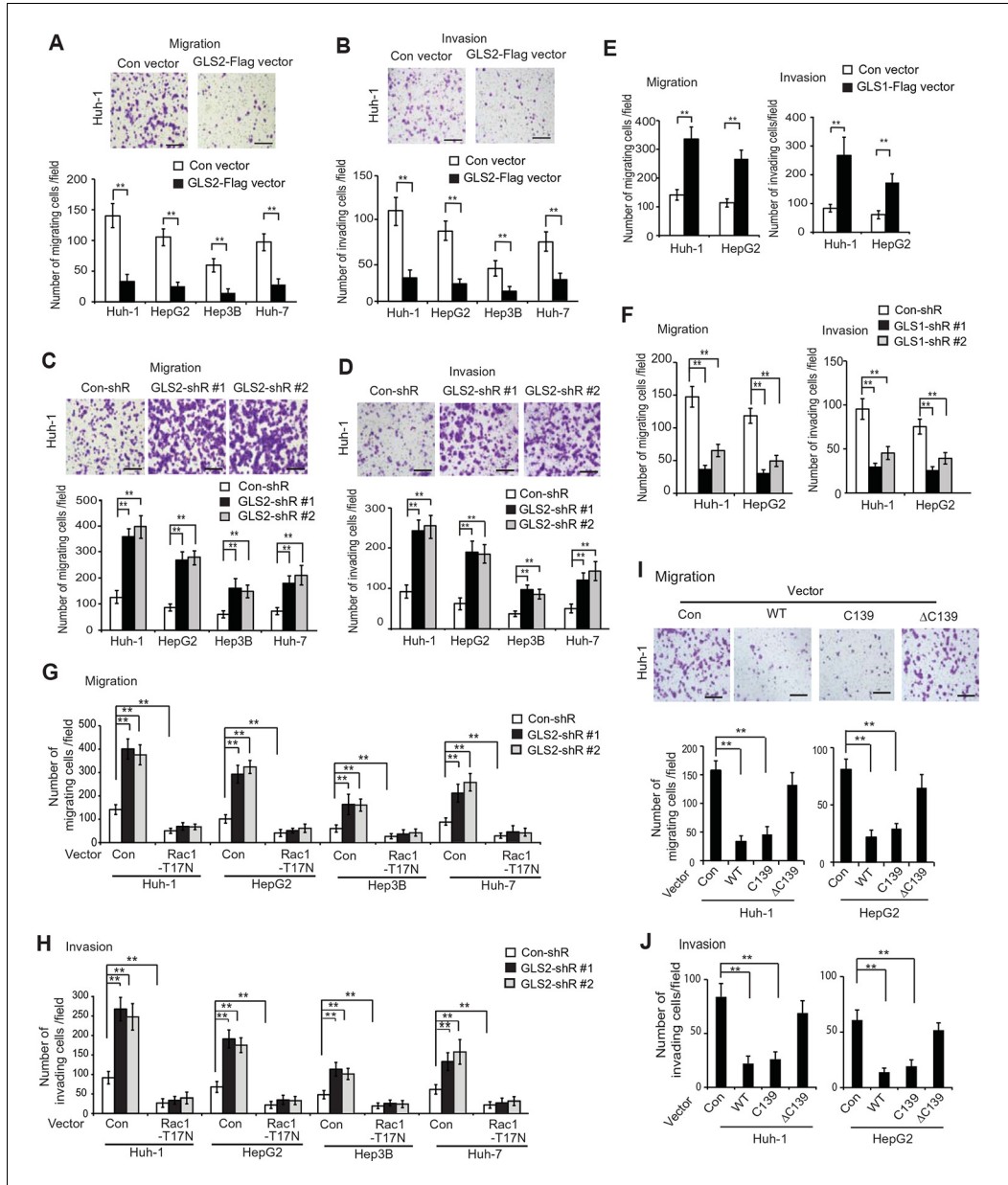

**Figure 4.** GLS2 inhibits migration and invasion of HCC cells through negative regulation of Rac1. (A, B) Ectopic GLS2 expression inhibited the migration (A) and invasion (B) of different HCC cells as determined by trans-well assays. Upper panels: representative images of migrating (A) and invading (B) Huh-1 cells transduced with control (con) or GLS2-Flag vectors. Scale bars: 200 μm. Lower panels: quantification of average number of migrating (A) and invading (B) cells/field in different HCC cells transduced with control (con) or GLS2-Flag vectors. Data present mean ± SD (n=6). **p<0.001; Student's t-test. (C, D) Knockdown of GLS2 by shRNA vectors promoted the migration (C) and invasion (D) of different HCC cells. Upper panels: representative images of migrating (C) and invading (D) Huh-1 cells with or without GLS2 knockdown. Scale bars, 200 μm. Data present mean ± SD (n=6). **p<0.001; Student's t-test. (E) Ectopic GLS1 expression promoted the migration (left) and invasion (right) of Huh-1 and HepG2 cells as determined by trans-well assays. (F) Knockdown of GLS1 decreased the migration (left) and invasion (right) of Huh-1 and HepG2 cells. In E, F, data are presented as mean ± SD (n=6). **p<0.001; Student's t-test. (G, H) Ectopic expression of DN Rac1-T17N largely abolished the promoting effects of GLS2 knockdown on migration (G) and invasion (H) of different HCC cells as measured by trans-well assays. Cells with knockdown of GLS2 by shRNA vectors were transduced with Rac1-T17N expression vectors for trans-well assays. Data present mean ± SD (n=6). **p<0.001; Student's t-test. (I, J) Ectopic expression of the C-terminus of GLS2, GLS2-C139, inhibited migration (I) and invasion (J) of Huh-1 and HepG2 cells as measured by trans-well assays. Cells were transduced with different GLS2 expression vectors described in *Figure 1E* for assays. Upper panels in I: representative images of migrating Huh-1 cells. Data present mean ± SD (n=6). **p<0.001; Student's t-test. GLS, glutaminase; HCC, hepatocellular carcinoma; shRNA, short hairpin RNA; WT, wild type.

*Figure 4 continued on next page*

*Figure 4 continued*

The following figure supplements are available for figure 4:

**Figure supplement 1.** Rac1 promotes the migration and invasion of Huh-1 and HepG2 cells.

**Figure supplement 2.** The viability and number of HCC cells with GLS2 overexpression or knockdown after being cultured in serum-free medium for 36 hr.

**Figure supplement 3.** Knockdown of endogenous Rac1 largely abolishes the effect of GLS2 on migration and invasion of HCC cells.

decreased expression of GLS2 in human HCC could be an important mechanism contributing to the high metastasis of human HCC. To this end, we investigated the association of decreased GLS2 expression with cancer metastasis in human HCC samples. We first queried the The Cancer Genome Atlas (TCGA) database to compare GLS2 expression between HCC samples with or without vascular invasion of HCC cells. As shown in *Figure 5G*, GLS2 expression was significantly lower in HCCs with vascular invasion (n=57), compared with HCCs without vascular invasion (n=110) (decreased by 3.03-fold; $p$=0.0066). Consistent results were also observed in another cohort from Gene Expression Omnibus (GEO, GSE6764) by using Oncomine, a human genetic dataset analysis tool. GLS2 expression was significantly lower in HCCs with vascular invasion (n=18), compared with HCCs without vascular invasion (n=15) (decreased by 4.62-fold; $p$=0.0198) (*Figure 5H*). These results indicated that the decreased GLS2 expression is significantly associated with enhanced metastasis in human HCC.

## GLS2 mediates p53's function in suppressing HCC metastasis

p53 plays a critical role in inhibiting cancer metastasis. However, while extensive work has been done on the mechanisms underlying p53-mediated apoptosis, cell cycle arrest and senescence, the mechanism underlying p53's function in suppressing cancer metastasis is much less well-understood (*Muller et al., 2011*; *Vousden and Prives, 2009*). Previous reports including ours have shown that as a direct p53 target, GLS2 is up-regulated by p53 in cells under both non-stressed and stressed conditions (*Hu et al., 2010*; *Suzuki et al., 2010*). Consistently, p53 knockdown by shRNA greatly reduced the mRNA and protein levels of GLS2 in Huh-1 and HepG2 cells which express WT p53 (*Figure 6A*). Considering the potent activity of GLS2 in inhibiting cancer cell metastasis, our findings raised the possibility that GLS2 may be an important mediator of p53's function in suppressing cancer metastasis.

Here, we tested this hypothesis. Knockdown of p53 significantly promoted the migration and invasion of both Huh-1 and HepG2 cells measured by trans-well assays (*Figure 6B,C*). As shown in *Figure 6—figure supplement 1*, no significant difference in the viability and number of these cells among different groups was observed after being cultured in serum-free medium for 36 hr at the end of trans-well assays. Notably, while individual knockdown of GLS2 or p53 dramatically promoted the migration and invasion of Huh-1 and HepG2 cells, simultaneous knockdown of GLS2 and p53 did not display a clear additive effect on the migration and invasion of these cells (*Figure 6D,E*). Consistently, while individual knockdown of GLS2 or p53 dramatically promoted lung metastasis of Huh-1 and HepG2 cells in mice, simultaneous knockdown of GLS2 and p53 did not display an additive effect on lung metastasis of these cells (*Figure 6F,G*). These results demonstrate that GLS2 is a novel and important mediator of p53 in suppressing cancer metastasis.

## GLS2 mediates p53's function in metastasis suppression through Rac1 inhibition

It has been reported that p53 inhibits Rac1 activity, but its mechanism remains unclear (*Bosco et al., 2010*; *Guo and Zheng, 2004*; *Muller et al., 2011*). As shown in *Figure 7A,B*, expression of DN Rac1-T17N greatly abolished the promoting effects of p53 knockdown on migration and invasion of Huh-1 and HepG2 cells. Consistently, Rac1 knockdown greatly abolished the promoting effects of p53 knockdown on migration and invasion of Huh-1 and HepG2 cells (*Figure 7—figure supplement 1A,B*). These results suggest that Rac1 inhibition is an important mechanism for p53 to inhibit metastasis.

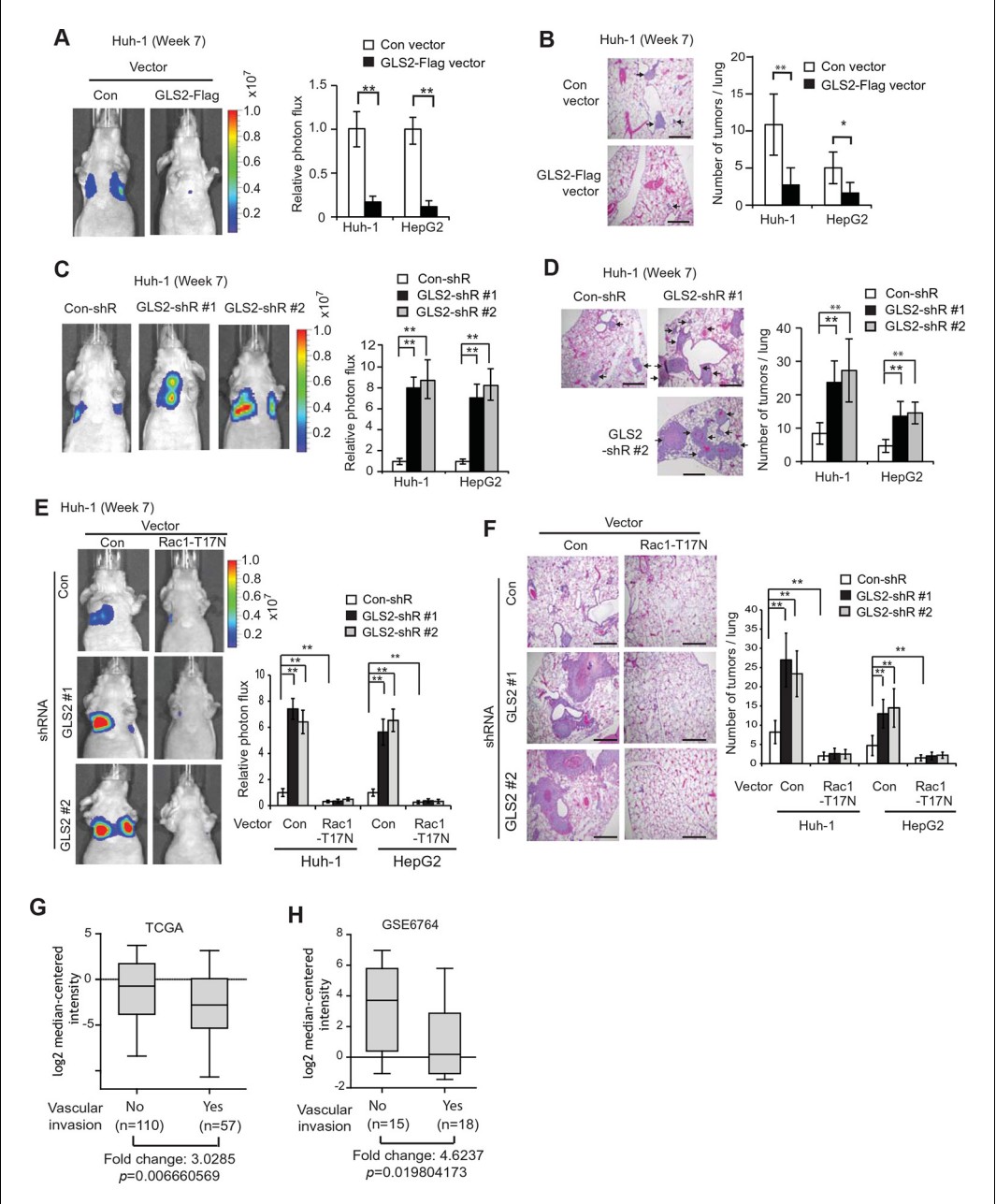

**Figure 5.** GLS2 inhibits lung metastasis of HCC cells in mice, and GLS2 expression is associated with metastasis in human HCC. (**A, B**) Ectopic expression of GLS2 inhibited lung metastasis of Huh-1 and HepG2 cells in nude mice. Huh-1 and HepG2 cells with GLS2 ectopic expression were transduced with lentiviral vectors expressing luciferase for lung metastasis assays. The lung metastasis was analyzed by in vivo bioluminescence imaging (**A**) and histological analysis (**B**) at 7 weeks after inoculation of cells. Left panels in A: representative images of lung metastasis of Huh-1 cells analyzed by in vivo imaging. Right panels in A: quantification of lung photon flux (photons per second). Left panels in B: representative images of hematoxylin and eosin staining of lung metastasis of Huh-1 cells. Right panels in B: The average number of tumors/lung. (**C, D**) Knockdown of endogenous GLS2 by shRNA promoted lung metastasis of Huh-1 and HepG2 cells in nude mice. (**E, F**) Ectopic expression of DN Rac1-T17N largely abolished the promoting effects of GLS2 knockdown on lung metastasis of HCC cells in vivo. Huh-1 and HepG2 cells with stable ectopic Rac1-T17N expression and GLS2 knockdown were used for lung metastasis assays in mice. In A-F, data represent mean ± SD (n=8 mice/group). *$p<0.01$; **$p<0.001$; Student's $t$-test. Scale bars, 200 μm. Arrows indicate metastatic tumors. (**G, H**) GLS2 expression is significantly decreased in human HCCs with metastasis compared with HCCs without metastasis. GLS2 mRNA expression in non-metastatic (without

*Figure 5 continued on next page*

*Figure 5 continued*

vascular invasion) and metastatic (with vascular invasion) HCCs was obtained from the TCGA (**G**) and GSE6764 (**H**). *p*=0.0066 in G; *p*=0.0198 in H; Student's *t*-test. GLS, glutaminase

Our finding that GLS2 interacts with Rac1-GDP to inhibit Rac1 activity suggests that as a direct p53 target, GLS2 could mediate p53's function in inhibiting Rac1 activity. Notably, while individual knockdown of GLS2 or p53 greatly activated Rac1, simultaneous knockdown of GLS2 and p53 did not display an additive effect on the Rac1 activity in Huh-1 or HepG2 cells (*Figure 7C*). Furthermore, GLS2-Flag overexpression largely abolished the promoting effect of p53 knockdown on the Rac1 activity in Huh-1 or HepG2 cells (*Figure 7D*). These results indicate that GLS2 mediates p53's function in inhibiting Rac1 activity.

Our results have shown that GLS2 inhibited the interaction between Rac1 and its GEFs Tiam1 and VAV1 to down-regulate the Rac1 activity (*Figure 3E–H*). Here, we investigated whether inhibition of the interaction of Tiam1 and VAV1 with Rac1 is an important mechanism for p53 to down-regulate the Rac1 activity. Knockdown of WT p53 in Huh-1 and HepG2 cells, which greatly decreased GLS2 protein levels (*Figure 7C*), clearly promoted the interaction of Tiam1 and VAV1 with Rac1 (*Figure 7E*). Notably, GLS2 knockdown in cells with p53 knockdown did not further promote the interaction of Tiam1 and VAV1 with Rac1 (*Figure 7E*). Furthermore, GLS2 overexpression largely abolished the promoting effect of p53 knockdown on the interaction of Tiam1 and VAV1 with Rac1 (*Figure 7F*). These results suggest that blocking the interaction of Tiam1 and VAV1 with Rac1-GDP by GLS2 contributes greatly to p53's function in inhibiting the Rac1 activity. Collectively, our results strongly suggest that GLS2 mediates p53's function in suppression of HCC metastasis by inhibiting the interaction of Rac1 GEFs, such as Tiam1 and VAV1, with Rac1-GDP to down-regulate the Rac1 activity (*Figure 7G*).

## Discussion

In this study, GLS2 was identified as a novel binding protein and negative regulator for Rac1. GLS2 bound to Rac1-GDP through its Switch I & II regions, which is also the binding domain for Rac1 GEFs Tiam1 and VAV1. Thus, GLS2 blocked the binding of Tiam1 and VAV1 to Rac1-GDP, and inhibited the Rac1 activation by Tiam1 and VAV1. We found that GLS2 inhibited migration and invasion of HCC cells in vitro and lung metastasis of HCC cells in vivo. Blocking the Rac1 signaling by expression of Rac1-T17N or knockdown of Rac1 largely abolished GLS2's function in inhibiting metastasis. These results demonstrate a novel and important role of GLS2 in suppressing cancer metastasis, and also reveal that GLS2 binding to Rac1-GDP to inhibit Rac1 activity is a critical underlying mechanism (*Figure 7G*).

GLS1 plays a critical role in promoting tumorigenesis through enhancing glutamine metabolism (*Gao et al., 2009*; *Thangavelu et al., 2012*; *Wang et al., 2010*). It is unclear why GLS1 and GLS2 have contrasting roles in tumorigenesis, although they both function as the glutaminase enzymes. While the glutaminase core domains of GLS1 and GLS2 show high homology, their C-termini show relatively low homology. In this study, we found that GLS2 bound to Rac1 through its C-terminus and inhibited the Rac1 activity to suppress migration and invasion of HCC cells. This effect requires the C-terminus of GLS2 and is independent of its glutaminase activity. In contrast, GLS1 does not bind to Rac1 or inhibit Rac1 activity. Considering the critical role of Rac1 in cancer, our results provide a novel mechanism for the different roles of GLS1 and GLS2 in tumorigenesis, particularly with respect to cancer metastasis. In addition to Rac1, GLS2 may interact with other proteins to regulate their functions, which in turn contributes to GLS2's function in tumor suppression. Future studies should shed light on the further mechanisms of GLS2 in tumor suppression.

p53 plays a critical role in suppressing cancer metastasis. While extensive work has been done on the mechanisms underlying p53-mediated apoptosis, cell cycle arrest and senescence, the mechanism underlying p53's function in suppressing cancer metastasis is much less well understood (*Muller et al., 2011*; *Vousden and Prives, 2009*). p53 has been reported to inhibit Rac1 activity, however, the detailed mechanism is unclear (*Bosco et al., 2010*; *Guo and Zheng, 2004*; *Muller et al., 2011*). Our results show that p53 knockdown down-regulated GLS2 levels and

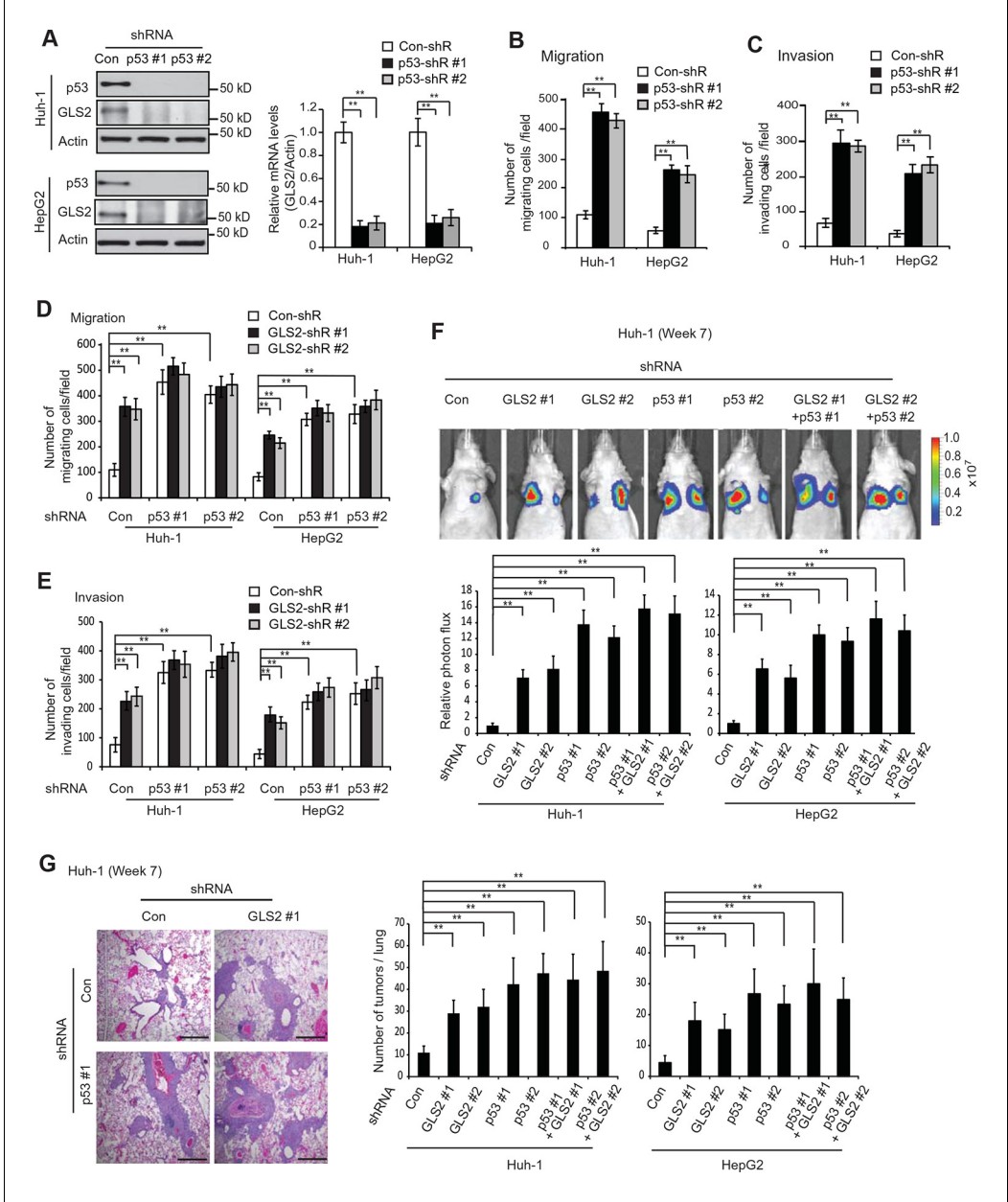

**Figure 6.** GLS2 mediates p53's function in inhibiting migration, invasion and lung metastasis of HCC cells. (**A**) Knockdown of endogenous WT p53 reduced GLS2 expression in Huh-1 and HepG2 cells as measured by western-blot (left) and Taqman real-time polymerase chain reaction assays (right), respectively. (**B, C**) Knockdown of p53 promoted the migration (**B**) and invasion (**C**) of Huh-1 and HepG2 cells measured by trans-well assays. (**D, E**) Simultaneous knockdown of GLS2 and p53 by shRNA vectors in Huh-1 and HepG2 cells did not display an addictive promoting effect on the migration (**D**) and invasion (**E**) of cells. In A–E, data represent mean ± SD (n=6). **$p<0.001$; Student's $t$-test. (**F, G**) Simultaneous knockdown of GLS2 and p53 in Huh-1 and HepG2 cells did not display an addictive promoting effect on lung metastasis in vivo. Huh-1 and HepG2 cells with individual knockdown of GLS2 or p53, or simultaneous knockdown of GLS2 and p53 were used for assays. In F, lung metastasis was analyzed by in vivo bioluminescence imaging at 7 weeks after inoculation of cells. Upper panels: representative images of lung metastasis of Huh-1 cells analyzed by in vivo imaging. Lower panels: quantification of lung photon flux. In G, lung metastasis was analyzed by histological analysis at week 7. Left panels: hematoxylin and eosin staining of lung metastasis of Huh-1 cells. Scale bars: 200 μm. Right panels: The average number of tumors/lung. Data represent mean ± SD (n=10 mice/group). **$p<0.001$; Student's $t$-test. GLS, glutaminase; HCC, hepatocellular carcinoma; shRNA, short hairpin RNA.

The following figure supplement is available for figure 6:

**Figure supplement 1.** The viability and number of HCC cells with p53 knockdown cultured in serum-free medium for 36 hr.

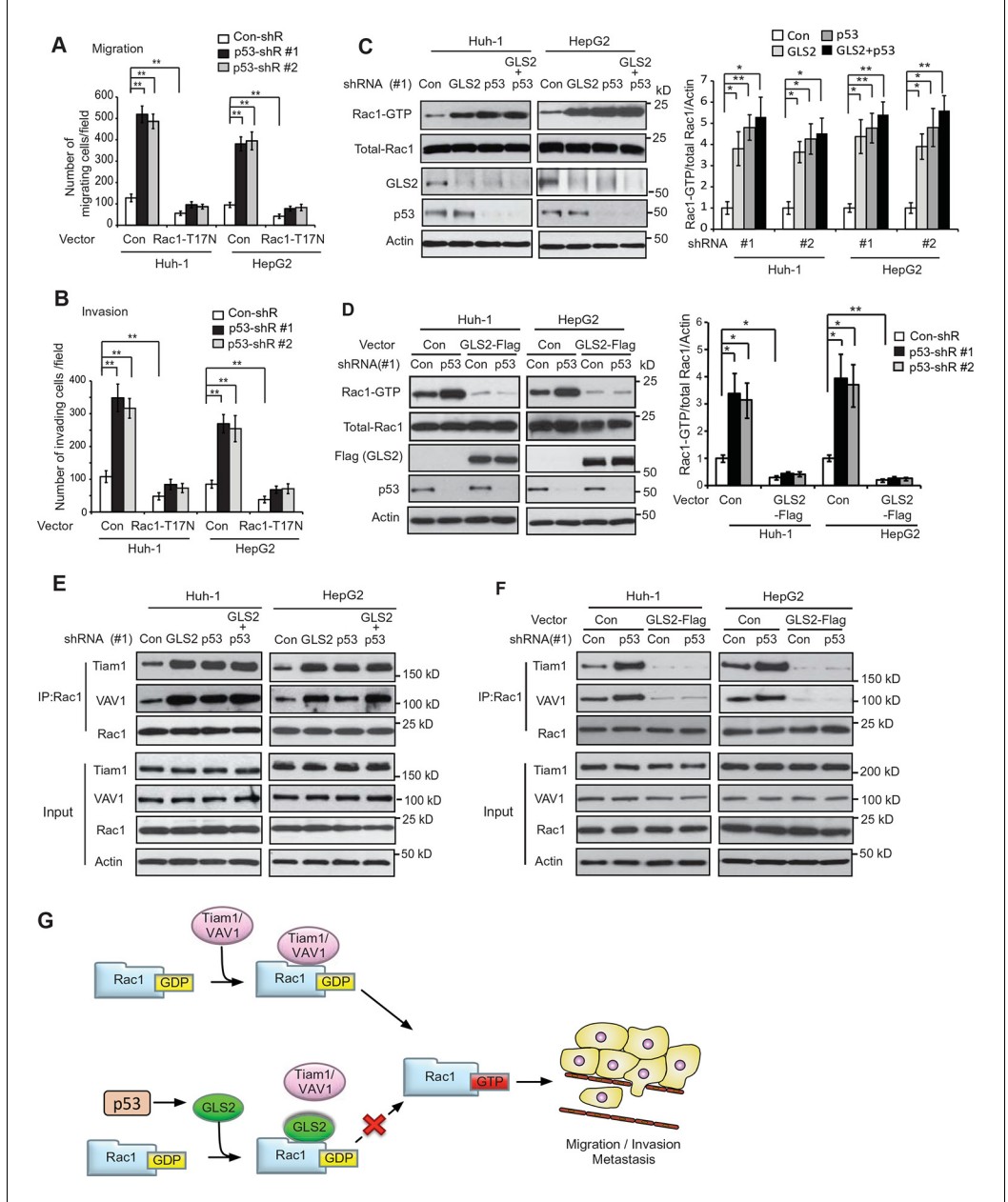

**Figure 7.** GLS2 mediates p53's function in negative regulation of the Rac1 activity. (**A, B**) Ectopic expression of DN Rac1-T17N greatly abolished the promoting effects of p53 knockdown on migration (**A**) and invasion (**B**) of Huh-1 or HepG2 cells as measured by trans-well assays. Data represent mean ± SD (n=6 in A, B). **$p<0.001$; Student's $t$-test. (**C, D**) GLS2 mediates p53's function in negative regulation of Rac1 activity in Huh-1 and HepG2 cells. In C, knockdown of p53 in cells with GLS2 knockdown did not further promote Rac1 activity. In D, GLS2 overexpression largely abolished the promoting effect of p53 knockdown on the Rac1 activity. Left panels: represented results of Rac1 activity analysis in cells transduced with #1 shRNA vectors. Right panel: relative Rac1-GTP/total Rac1/Actin levels in cells transduced with two different shRNA vectors (#1 and #2). Data represent mean ± SD (n=3). *$p<0.01$; **$p<0.001$; Student's $t$-test. (**E, F**) p53 inhibits the interaction of Rac1 with Tiam1 and VAV1 through GLS2 in Huh-1 and HepG2 cells. In E, knockdown of p53 in cells with GLS2 knockdown did not further promote the interaction of Tiam1 and VAV1 with Rac1. The knockdown of p53 and GLS2 was shown in *Figure 7C*. Two shRNA vectors against p53 and GLS2, respectively, were used, and very similar results were observed. In F, GLS2 overexpression largely abolished the promoting effect of p53 knockdown on the interaction of Tiam1 and VAV1 with Rac1. (**G**) Proposed model for the negative regulation of Rac1 activity and cancer metastasis by GLS2 and p53. GDP, guanosine 5'-diphosphate; GLS, glutaminase; GTP, guanosine 5'-triphosphate; shRNA, short hairpin RNA.

The following figure supplement is available for figure 7:

**Figure supplement 1.** Knockdown of endogenous Rac1 greatly abolished the effects of p53 on migration and invasion of HCC cells.

promoted the interaction of Rac1-GDP with its GEFs Tiam1 and VAV1, and thereby enhanced the Rac1 activity and promoted cancer metastasis. These results demonstrate that GLS2 is a novel and critical mediator for p53 in suppressing metastasis, and also reveal a novel mechanism by which p53 inhibits Rac1.

HCC is one of the most common types of cancer and the third leading cause of cancer death worldwide (*El-Serag and Rudolph, 2007*; *Jemal et al., 2011*). The high malignancy and poor prognosis of HCC has been related with the high metastatic characteristic of HCC; however, the mechanism underlying HCC metastasis is not well understood (*El-Serag and Rudolph, 2007*; *Tang, 2001*). Considering that GLS2 is frequently down-regulated in HCC (*Hu et al., 2010*; *Liu et al., 2014a*; *Suzuki et al., 2010*; *Xiang et al., 2015*), our findings that GLS2 inhibits HCC metastasis and loss of GLS2 promotes HCC metastasis provide a novel mechanism contributing to high metastatic characteristic of HCC. These results strongly suggest that the GLS2/Rac1 signaling could be a potential target for therapy in cancer, particularly in HCC.

In summary, our results demonstrate that GLS2 is a novel negative regulator of Rac1, and plays a novel and critical role in suppression of metastasis through its negative regulation of the Rac1 activity. Furthermore, our results also reveal that GLS2 is a critical mediator for p53 in suppression of cancer metastasis.

## Materials and methods

### Cell lines, vectors and shRNA

HepG2 (p53-WT) and Hep3B (p53-null) cells were obtained from American Type Culture Collection (ATCC, Manassas, VA). Huh-1 (p53-WT) and Huh-7 (p53 mutant) were obtained from the Japanese Culture Collection (RIKEN BioResource Center, Saitama, Japan). All cell lines were authenticated by short tandem repeat profiling. Cells were regularly tested for mycoplasma using Lookout Mycoplasma PCR detection kit (MP0035, Sigma, St. Louis, MO) and only used when negative. The WT p53 were knocked down in HepG2 and Huh-1 cells by 2 different shRNA vectors as previously described (*Zhang et al., 2013*). The pLPCX vectors expressing Flag-tagged WT or deletion mutants of GLS2 were constructed by PCR amplification as we previously described (*Hu et al., 2010*). The pLPCX-Myc-Rac1 and pLPCX-Myc-Rac1-G12V vectors were constructed by using Myc-Rac1 DNA fragment from pcDNA3.1-Myc-Rac1 WT and pcDNA3.1-Myc-Rac1 G12V vectors, respectively (*He et al., 2010*). The pLPCX-Myc-Rac1-T17N vector was constructed by using a Quikchange II XL Site-Directed Mutagenesis Kit (Stratagene/Agilent Technologies, San Diego, CA). The pLPCX vectors expressing deletion mutants of Myc-Rac1 were constructed by PCR amplification. The pLPCX-GLS1-Flag, pLPCX-Tiam1-HA and pLPCX-VAV1-HA vectors were cloned using PCR amplification. Two lentiviral shRNA vectors against GLS2 (ID: V3LHS_307701 and V2LHS_71048), two lentiviral shRNA vectors against Rac1 (ID: V3LHS_317664 and V3LHS_317668) and control shRNA vectors were obtained from Open Biosystems (Huntsville, AL).

### Cell migration and invasion assays

The trans-well system (24 wells, 8 µM pore size, BD Biosciences, Franklin Lakes, NJ) was employed for cell migration and invasion assays as we previously described (*Zheng et al., 2013*; *Zhao et al., 2015*). In brief, cells ($2 \times 10^4$ for Huh-1, and $6 \times 10^4$ for HepG2) in serum-free medium were seeded into upper chambers for migration assays. Cells on the lower surface were fixed, stained and counted at 24 hr after seeding. For invasion assays, cells ($4 \times 10^4$ for Huh-1, and $1 \times 10^5$ for HepG2) were seeded into upper chambers coated with matrigel (BD Biosciences). Cells on the lower surface were fixed, stained and counted at 36 hr after seeding.

### In vivo lung metastasis assays

In vivo lung metastasis assays were performed as previously described (*Li et al., 2014*; *Zheng et al., 2013*). In brief, Huh-1 and HepG2 cells ($2 \times 10^6$ in 0.1 mL phosphate-buffered saline) transduced with lentiviral vectors expressing luciferase were injected into 2-month-old male BALB/c nude mice *via* the tail vein (n=8 mice/group). Lung metastatic colonization was monitored and quantified at different weeks using non-invasion bioluminescence imaging by IVIS Spectrum in vivo imaging system (PerkinElmer, Waltham, MA), and was validated at the endpoint by routine histopathological

analysis. All mouse experiments were approved by the University Institutional Animal Care and Use Committee.

## Western blot assays

Standard western blot assays were used to analyze protein expression in cells. The following antibodies were used for assays: anti-Flag-M2 (F1804, Sigma; 1:20,000 dilution), anti-β-Actin (A5441, Sigma; 1:10,000 dilution), anti-Myc (9E10, Roche, Indianapolis, IN; 1:1000 dilution), anti-HA (3F10, Roche; 1:1000 dilution), anti-Rac1 (23A8, Millipore, Billerica, MA; 1:5000 dilution), anti-p-PAK (Ser199/204) (09–258, Millipore; 1:1000 dilution), anti-PAK (07–1451, Millipore; 1: 1000 dilution), anti-p53 (FL393, Santa Cruz, Dallas, TX; 1:2000 dilution), anti-Tiam1 (sc-872, Santa Cruz; 1:2000 dilution), anti-VAV1 (sc-8039, Santa Cruz; 1:1000 dilution). The anti-GLS2 antibody (1: 1000 dilution) was prepared as previously described (*Hu et al., 2010*). To increase the sensitivity of the GLS2 antibody, endogenous GLS2 in cells was pulled down by IP and detected by western blot assays. The band intensity was quantified by digitalization of the X-ray film and analyzed with the ImageJ software.

## Co-IP assays

Co-IP assays were performed as we previously described (*Liu et al., 2014b*). For co-IP of GLS2-Flag and Myc-Rac1 proteins, anti-Flag (M2, Sigma) and anti-Myc (9E10, Roche) agarose beads were used to pull down GLS2-Flag and Myc-Rac1, respectively. For Co-IP of endogenous GLS2 and Rac1, the anti-GLS2 antibody and the anti-Rac1 (23A8, Millipore) antibody were used for IP, respectively. The mouse or rabbit purified IgGs were used as negative controls.

## LC-MS/MS analysis

To determine potential GLS2 binding proteins, GLS2-Flag protein in Huh-1 cells with stable expression of GLS2-Flag was pulled down by co-IP using anti-Flag (M2) beads and eluted with Flag peptide. Huh-1 cells transduced with control vectors were used as a control for co-IP assays. Eluted materials were separated in a 4–16% sodium dodecyl sulfate polyacrylamide gel electrophoresis and visualized by silver staining using the silver staining kit (Invitrogen, Carlsbad, CA). LC-MS/MS analysis was performed at the Biological Mass Spectrometry facility of Rutgers University as previously described (*Yue et al., 2015*; *Zhao et al., 2015*).

## Rac1 activity analysis

For Rac1 activity analysis, the GST-p21-binding domain of PAK1 pull-down assays were performed using a Rac1 activation assay kit (Millipore) to measure the levels of GTP-bound Rac1 (Rac1-GTP) in cells as previously described (*Galic et al., 2014*; *Hayashi-Takagi et al., 2010*; *Palacios et al., 2002*). The p21-binding domain of the Rac1 effector protein PAK1 binds specifically to the Rac1-GTP (*Hayashi-Takagi et al., 2010*; *Palacios et al., 2002*). The levels of precipitated Rac1-GTP were measured by western blot assays using a Rac1 antibody (23A8, Millipore) and normalized to total Rac1 levels in cells measured by western-blot assays.

## Quantitative real-time PCR assays

Total RNA was prepared with the RNeasy Kit (Qiagen, Hilden, Germany). Complementary DNA was prepared using a TaqMan reverse transcription kit, and real-time PCR was performed with TaqMan PCR mixture (Applied Biosystems, Foster City, CA) as we previously described (*Hu et al., 2010*; *Zhang et al., 2013*). The expression of genes in cells was normalized to the expression of the *Actin* gene.

## Glutaminase activity assays

Glutaminase activity was measured as previously described (*Gao et al., 2009*; *Wang et al., 2010*). Briefly, cell lysates were incubated at 37°C for 10 min with the assay mix consisting of 20 mM glutamine, 50 mM Tris-acetate (pH 8.6), 100 mM phosphate, and 0.2 mM ethylenediaminetetraacetic acid. The reaction was quenched with the addition of 2 mL of 3 M HCl. Subsequently, the reaction mixture was incubated for 30 min at room temperature with the second assay mix (2.2 U glutamate dehydrogenase, 80 mM Tris-acetate (pH 9.4), 200 mM hydrazine, 0.25 mM ADP, and 2 mM nicotinamide adenine dinucleotide). The absorbance was read at 340 nm using a spectrophotometer.

## Statistical analysis

The differences in tumor growth among groups were analyzed for statistical significance by analysis of variance, followed by Student's *t*-tests using GraphPad Prism software. All other *P*-values were obtained using two-tailed Student *t*-tests. \*\**p*<0.001; \**p*<0.01; #*p*<0.05.

## Acknowledgements

We thank Dr. Arnold Levine for helpful discussion and comments. This work was supported by grants from the NIH (1R01CA143204), CINJ Foundation and New Jersey Health Foundation (to ZF), by grants from NIH (1R01CA160558-0 1) and Ellison Medical Foundation (to WH), by grants from NIH (R01CA169182-01 to GB), and BCRF (to BGH). YuZ was supported by China Scholarship Council (#201406320151). JL was supported by NJCCR postdoctoral fellowship. This research was supported by Preclinical Imaging Shared Resource of Rutgers Cancer Insitute of New Jersey (P30CA072720).

## Additional information

### Funding

| Funder | Grant reference number | Author |
| --- | --- | --- |
| National Institutes of Health | 1R01CA143204 | Zhaohui Feng |
| National Institutes of Health | 1R01CA160558 | Wenwei Hu |
| National Institutes of Health | R01CA169182-01 | Gyan Bhanot |

The funders had no role in study design, data collection and interpretation, or the decision to submit the work for publication.

### Author contributions

CZ, JL, BGH, ZF, Conception and design, Acquisition of data, Analysis and interpretation of data, Drafting or revising the article; YZ, XY, YZ, XW, HW, FB, SL, Acquisition of data, Analysis and interpretation of data, Drafting or revising the article; GB, WH, Analysis and interpretation of data, Drafting or revising the article, Contributed unpublished essential data or reagents

### Ethics

Animal experimentation: This study was performed in strict accordance with the recommendations in the Guide for the Care and Use of Laboratory Animals of the National Institutes of Health. All of the animals were handled according to approved institutional animal care and use committee (IACUC) protocol (I12-002) of Rutgers, State University of New Jersey.

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
