## [Decision Letter]

Thank you for submitting your work entitled "Glutaminase 2 is a novel negative regulator of Rac1 and mediates p53 function in suppressing cancer metastasis" for peer review at *eLife*. Your submission has been favorably evaluated by Charles Sawyers (Senior editor), Michael Green (Reviewing editor), and three expert reviewers.

The reviewers have discussed the reviews with one another and the Reviewing editor has drafted this decision to help you prepare a revised submission.

Summary:

Increased glutaminolysis, the catabolism of glutamine into TCA cycle intermediates, is recognized as a key metabolic alteration which occurs in cancer cells. An emerging question is why the glutaminase isozymes GLS1 and GLS2, despite being enzymatically equivalent, appear to serve opposite roles in tumorigenesis, with recent studies showing a tumor suppressive activity of GLS2.

Here, the authors provide an explanation for this – they show that GLS2, but not GLS1, binds to and inhibits Rac1. This is done independently of enzymatic activity, via the binding of GLS2 to the switch regions of Rac1, preventing the binding of Rac1 GEFs. Through a series of gain-of-function and loss-of-function based experiments in multiple cell lines and in vivo, the authors show that this mechanism is responsible for the migration/metastasis inhibiting activities of GLS2. Furthermore, as GLS2 has been reported to be a target of p53, the authors show that GLS2 inhibition of Rac1 mediates in large part the known suppressive properties of p53 on migration and metastasis.

The question on hand is highly relevant to understanding the regulation of cancer cell metastasis, and to understanding the role of the glutaminases in cancer. While the role in cancer of the genes addressed in the study were already known, the core mechanism of the study, the inhibitory role of GLS2 on Rac1, is novel, well detailed, and nicely ties together previous knowledge of p53/GLS2/Rac1 in metastasis. Overall, experiments are well designed and very well executed, with convincing results. There are a few concerns and issues that need to be addressed in order to strengthen the study. They are listed in order of importance below:

Essential revisions:

1) Despite the previous studies on GLS2 showing its negative effects on cell proliferation/viability, the authors only focus on the migration/metastasis aspect here. This is an important and obvious missing control since decreased cell viability can indirectly impact migration/invasion/metastasis. Is GLS2 inhibition of Rac1 involved in the previously reported effects of GLS2 on cell survival? In the effects of p53 on survival? While it is unlikely that the results on cell migration/metastasis are artifacts of effects on cell viability, especially in light of the increased migration following GLS2 loss, cell viability for a few key conditions need to be examined and discussed.

2) In Figure 7, the lack of an additive effect of knocking down p53 and GLS2 suggests but does not prove that GLS2 mediates p53's function in inhibiting Rac1. A more direct experiment to verify this would be to see if overexpression of GLS2 nullifies the effects of p53 knockdown on Rac1 activation and Tiam1/Vav1 binding.

3) Endogenous expression levels of GLS1 and GLS2 expression levels for the four HCC cell lines used in the study should be shown. Also, what is the degree of overexpression of GLS2 or GLS2 fragments in the experiments, relative to endogenous GLS2?

---

## [Author Response]

Essential revisions:

*1) Despite the previous studies on GLS2 showing its negative effects on cell proliferation/viability, the authors only focus on the migration/metastasis aspect here. This is an important and obvious missing control since decreased cell viability can indirectly impact migration/invasion/metastasis. Is GLS2 inhibition of Rac1 involved in the previously reported effects of GLS2 on cell survival? In the effects of p53 on survival? While it is unlikely that the results on cell migration/metastasis are artifacts of effects on cell viability, especially in light of the increased migration following GLS2 loss, cell viability for a few key conditions need to be examined and discussed.*

Thanks for raising this important question. Recent reports including ours show that GLS2 negatively regulates cell proliferation (Hu et al., PNAS, 2010; Suzuki et al., PNAS, 2010). Results from our recent study suggest that inhibition of AKT signaling is an important mechanism contributing to the inhibitory effect of GLS2 on cell proliferation (Liu et al., Oncotarget, 2014). In the current study, we found that GLS2 inhibits Rac1 activation. Rac1 activation promotes cell proliferation. It is therefore possible that the inhibition of Rac1 by GLS2 is an additional mechanism contributing to the role of GLS2 in negative regulation of cell proliferation, which is worth further investigation in a future study. In this study, we focused the effect of GLS2 on migration, invasion and metastasis of HCC cells. Therefore, a serum-free medium was used in in vitrotrans-well assays to determine the abilities of migration and invasion of cells with ectopic expression of GLS2, knockdown of endogenous GLS2 or their control cells. Serum starvation induces G0/G1 cell cycler arrest and inhibits cell proliferation. Under this experiment condition for 36 h (the length for migration and invasion transwell assays), GLS2 (either ectopic expression or knockdown of GLS2) does not have an obvious effect on cell proliferation as well as cell viability (Figure 4—figure supplement 2). These results strongly suggest that the effect of GLS2 on migration and invasion is not due to the effect of GLS2 on cell viability or proliferation. As suggested, we have also added Figure 4—figure supplement 3 and Figure 6—figure supplement 1 as controls to show that no significant difference in the viability and number of HCC cells among different groups was observed after being cultured in a serum-free medium for 36 h at the end of trans-well assays in HCC cells with Rac1 or p53 knockdown.

*2) In Figure 7, the lack of an additive effect of knocking down p53 and GLS2 suggests but does not prove that GLS2 mediates p53's function in inhibiting Rac1. A more direct experiment to verify this would be to see if overexpression of GLS2 nullifies the effects of p53 knockdown on Rac1 activation and Tiam1/Vav1 binding.*

Thanks for this good suggestion. We did the experiments as suggested, and found that overexpression of GLS2 largely abolished the effects of p53 knockdown on Rac1 activation (Figure 7) and Tiam1/Vav1 binding (Figure 7). These data are consistent with our results showing the lack of an additive effect of knocking down p53 and GLS2 on Rac1 activation (Figure 7) and Tiam1/Vav1 binding (Figure 7). These results together strongly suggest that GLS2 mediates p53's function in inhibiting Rac1. These new data have been added to Figure 7 and Figure 7.

*3) Endogenous expression levels of GLS1 and GLS2 expression levels for the four HCC cell lines used in the study should be shown. Also, what is the degree of overexpression of GLS2 or GLS2 fragments in the experiments, relative to endogenous GLS2?*

As suggested, we presented the endogenous GLS1 and GLS2 protein levels in different HCC cells in Figure 2—figure supplement 2.

Due to relatively narrow quantitative linear range of western-blot assays, we measured the mRNA levels of endogenous GLS2 and exogenous GLS2 in HCC cells by Taqman real-time PCR assays. We also included two normal liver tissues as controls. Compared with normal liver tissues, the mRNA level of endogenous GLS2 is greatly reduced (by around 10-20-fold) in HCC cells, and the exogenous GLS2 in HCC cells with ectopic GLS2 expression is around two-three fold higher than normal liver tissues (Figure 2—figure supplement 2).

The GLS2 fragments are mutant GLS2 with deletion of different domains, it is difficult to detect their mRNA levels in cells with the same sets of primers for real-time PCR assays and directly compare their expression with the levels of endogenous GLS2 in cells. As shown in Figure 1 and Figure 2, the protein levels of these Flag-tagged GLS2 fragments are similar to the levels of the wild-type GLS2-Flag in cells as measured by western-blot assays with the anti-Flag antibody. According to the results in Figure 2—figure supplement 2, the mRNA expression levels of these different GLS2 fragments in cells should be similar to that of WT GLS2-Flag, which is around two-three-fold higher than endogenous GLS2 in normal tissues, and 20-60-fold higher than the endogenous GLS2 in these HCC cells (Figure 2—figure supplement 2).